# Delivery and Transcriptome Assessment of an In Vitro Three-Dimensional Proximal Tubule Model Established by Human Kidney 2 Cells in Clinical Gelatin Sponges

**DOI:** 10.3390/ijms242115547

**Published:** 2023-10-24

**Authors:** Hui-Yi Hsiao, Tzung-Hai Yen, Fang-Yu Wu, Chao-Min Cheng, Jia-Wei Liu, Yu-Ting Fan, Jung-Ju Huang, Chung-Yi Nien

**Affiliations:** 1Department of Biomedical Sciences, College of Medicine, Chang Gung University, Taoyuan 33302, Taiwan; ivyhsiao@gmail.com; 2Center for Tissue Engineering, Linkuo Chang Gung Memorial Hospital, Taoyuan 33305, Taiwan; jiaweiliu1016@gmail.com; 3Department of Nephrology, Clinical Poison Center, Linkuo Chang Gung Memorial Hospital, Taoyuan 33305, Taiwan; m19570@adm.cgmh.org.tw; 4Department of Nephrology, College of Medicine, Chang Gung University, Taoyuan 33302, Taiwan; 5Department of Life Science, National Central University, Taoyuan 32001, Taiwan; wfu850920@gmail.com (F.-Y.W.); amanda89718@gmail.com (Y.-T.F.); 6Institute of Biomedical Engineering, National Tsing Hua University, Hsinchu 300193, Taiwan; chaomin@mx.nthu.edu.tw; 7Division of Reconstructive Microsurgery, Department of Plastic and Reconstructive Surgery, Linkou Chang Gung Memorial Hospital, Taoyuan 33305, Taiwan; jungjuhuang@gmail.com

**Keywords:** proximal tubule cells, two-dimensional culture, three-dimensional culture, HK-2 cells, gelatin spongy scaffold, comparative transcriptome analysis, nephrotoxicity

## Abstract

The high prevalence of kidney diseases and the low identification rate of drug nephrotoxicity in preclinical studies reinforce the need for representative yet feasible renal models. Although in vitro cell-based models utilizing renal proximal tubules are widely used for kidney research, many proximal tubule cell (PTC) lines have been indicated to be less sensitive to nephrotoxins, mainly due to altered expression of transporters under a two-dimensional culture (2D) environment. Here, we selected HK-2 cells to establish a simplified three-dimensional (3D) model using gelatin sponges as scaffolds. In addition to cell viability and morphology, we conducted a comprehensive transcriptome comparison and correlation analysis of 2D and 3D cultured HK-2 cells to native human PTCs. Our 3D model displayed stable and long-term growth with a tubule-like morphology and demonstrated a more comparable gene expression profile to native human PTCs compared to the 2D model. Many missing or low expressions of major genes involved in PTC transport and metabolic processes were restored, which is crucial for successful nephrotoxicity prediction. Consequently, we established a cost-effective yet more representative model for in vivo PTC studies and presented a comprehensive transcriptome analysis for the systematic characterization of PTC lines.

## 1. Introduction

The kidney plays important roles in maintaining homeostasis and eliminating body wastes, drugs, and toxins. Kidney diseases have become a major global health issue and are among the leading causes of disease burden, stemming from various factors such as diabetes, hypertension, genetic effects, drug-induced injuries, or environmental exposures. The proximal tubule, a segment of nephrons, is responsible for the major reabsorption of the glomerular ultrafiltrate and secretion of various solutes and compounds from the circulation [1,2]. Unlike glomerular filtration, tubular transportation is selective and directional via three systems: receptor-mediated endocytosis, transporter-mediated reabsorption, and transporter-mediated secretion. Renal proximal tubular cells (PTCs) exhibit high metabolic activity and contain a variety of carriers and transporters differentially localized on the apical or basolateral membranes [3]. Through the complex work of drug transporters, the PTCs are a major route to excrete drugs and xenobiotics [4,5].

The roles of PTCs make them particularly vulnerable to nephrotoxicity and important tools for nephrotoxicity prediction, drug development, and pathophysiological research [6,7,8]. The most common approach still relies on in vitro two-dimensional (2D) cell culture due to its simplicity, cost-effectiveness, repeatability, and fewer ethical concerns [9]. However, the limitations of 2D culture, which fails to reproduce in vivo tissue complexity, have been increasingly recognized [10,11]. Many proximal tubule cell (PTC) lines are available but have been indicated to be less sensitive to known nephrotoxins, mainly due to altered expression of transporters in the 2D environment [12]. In fact, Khundmiri et al. recently compared the expression profile of 14 frequently researched renal cell lines from six species to that of native mouse kidney proximal tubules [13,14,15]. None of them fully matched native mouse PTCs. Notably, relatively lower percentage matches were observed for human lines, such as human kidney 2 (HK-2) cells, decreasing their potential use for nephrotoxicity evaluation and prediction. However, a direct comparison to native human PTCs has not yet been performed.

Many efforts have been made to improve in vitro models with more physiological relevance, including three-dimensional (3D) cultures or more sophisticated models, such as iPSC-derived organoids or organ-on-a-chip (OOC) systems [16,17]. These models better maintain the morphology, polarity, gene expression, and metabolism closer to in vivo cells [9,16,18]. For example, an immortalized human renal proximal tubule epithelial cell (RPTEC/TERT1) line, which lacks functional organic anion transporters (OATs) in 2D conditions, was cultured in a 3D Matrigel sandwich. A differentiated tubule-like structure developed after two weeks of culture with exclusive expression of OAT3 [19]. Another example is that the Nki-2 cell line, derived from hTERT immortalized human renal cortical cells, was encapsulated in a mixture of Matrigel and collagen I. Tube formation was observed after two weeks, accompanied by exclusive expression of megalin for endocytosis [20] and the secretion of kidney injury molecule-1 (Kim-1), a biomarker for kidney injury [21], presenting higher sensitivity to cisplatin. Despite significant improvements observed in these models, certain challenges, such as higher costs, complicated techniques, difficulties in imaging and extracting cells, lower data consistency and reproducibility, and the source of kidney tissues, have been holding scientists back from fully adopting the 3D model or emerging organoid and OOC system, especially for high-throughput studies [22,23]. It is essential to establish a feasible 3D kidney model that is simple and practical for standardized high-throughput applications. Meanwhile, a more systematic characterization and comprehensive transcriptome assessment of the outcomes are also needed.

In this study, HK-2 cells were selected to establish a simplified 3D model using clinical-grade gelatin sponges as scaffolds and subjected to comparative transcriptome analysis. The immortalized HK-2 cell line was generated from human renal-cortex-derived cells [24]. Although 2D cultured HK-2 cells have been extensively used in nephrotoxicity studies [25,26], limited expression of certain key transporters has been reported, resulting in poor performance in nephrotoxicity prediction [27,28]. Distinctive cell behaviors were observed when using sponge or hydrogel scaffolds for cartilage regeneration. Zhang et al. discovered that chondrocytes exhibited more prominent proliferation and chondrogenic gene expression when cultured in a sponge scaffold, which was attributed to internal structural differences [29]. By providing a 3D microenvironment with more interconnective architecture, we investigated whether HK-2 cells could better mimic in vivo PTC.

In addition to cell viability and morphology evaluation, we conducted a comprehensive transcriptome comparison and correlation analysis between HK-2 cells cultured in 2D and 3D conditions and native human PTCs [30]. The gelatin-based 3D HK-2 model displayed stable and long-term growth with a tubule-like morphology and expression of major genes involved in the transport and metabolic processes of PTCs, crucial for successful nephrotoxicity prediction. It holds great promise for establishing a cost-effective yet more accurate renal model for enhanced nephrotoxicity studies and better translational research in kidney-related pathophysiology.

## 2. Results

### 2.1. Growth Pattern and Morphology of HK-2 Cells in the 2D and 3D Culture

To monitor the proliferation and morphological differences in HK-2 cells in 2D and 3D environments, 8000 or 80,000 HK-2 cells were seeded onto commercial gelatin sponge disks (8 mm in diameter and 1 mm in height) for the 3D model, as well as for their respective 2D model. The growth of HK-2 cells in both conditions was evaluated by the WST-8 assay (Figure 1A,B) and the live and dead assay (Figure 1C–H) at different time points. The growth curves of the 3D groups with both seeding numbers showed a similar pattern, which initially grew slower than their counterparts in the 2D culture but subsequently surpassed the growth rate of the 2D groups after 7 days of culture. Rapid and linear growth was observed from Day 5 until reaching a plateau on Day 17 and slightly decreased on Day 21. The 3D group with a higher seeding number (80 K) showed an enhanced but not proportional proliferation rate compared to the group with a lower seeding number (8 K), possibly due to the constraints of the sponge size. In the 2D culture, cells showed a fluctuating growing curve. The overall capacity of HK-2 cell proliferation was significantly higher in the 3D condition than in the 2D condition.

The live and death staining results (Figure 1C–F) and quantification (Figure 1G,H) of the 3D cultured cells were consistent with the WST-8 assay. The majority of the cells survived, and the numbers gradually increased as the culture period extended. Overall, the proliferation of 3D cultured cells displayed long-term and stable growth. In 2D culture, cells with low and high seeding numbers reached their peaks on Day 14 and Day 7, respectively, and maintained similar numbers of live cells throughout the culture period. Meanwhile, a high percentage of dead cells was detected after Day 14. Repetitive processes of empty regions due to cell death and refilled regions of proliferation were observed (Appendix A). This suggested an uneven and unstable cell growth in 2D conditions after Day 7, reflected by the fluctuating curve observed in the WST-8 assay.

Morphologically, 3D cultured cells started to form tubule-like structures on Day 7 (Figure 1J,K) and became more prominent on Day 21 (Figure 1E,F). This structure was absent in the 2D culture (Figure 1C,D,I). Instead, the cells exhibited varying sizes and shapes and became aggregated with dome-shaped colonies after Day 7 (Appendix A). The morphological differences within 2D and 3D conditions suggested that the gelatin sponge provided an architecture more closely resembling the in vivo environment, facilitating stable proliferation and well-differentiated morphology.

### 2.2. Comparative Transcriptomic Analysis of HK-2 Cells and Human Native PTCs

Gene expression profiling of HK-2 cells cultured in 3D gelatin sponges was performed by RNA-seq and compared to two reported datasets from 2D cultured HK-2 cells [13,31] and an additional dataset of native human PTCs obtained from single-cell RNA-seq (scRNA-seq) [30]. The sources and culture conditions of each dataset used in this study are listed in Table 1.

To determine expression differences associated with culture conditions, differentially expressed genes (DEGs) between 3D and each 2D group were analyzed and highlighted using volcano plots (Figure 2A,B). In comparison to the 2DHK2_K group, 677 genes were upregulated, and 264 genes were downregulated within the 3D group. The number of DEGs was much greater, exceeding 2000 genes for both up- and downregulated genes, when compared to the 2DHK2_Z group, suggesting more pronounced expression differences. This was further supported by the principal component analysis (PCA) result based on whole-genome expression (Figure 2C). All three replicates of the 3D groups (3DHK2) clustered together and were close to the 2DHK2_K groups but distinctively separated from the 2DHK2-Z group, likely due to the difference in culture medium. 

To investigate the representativeness of the 3D model to in vivo human PTCs, the single-cell expression profiling of 23,366 human kidney cells [30] was also incorporated into the PCA. Following the previous analysis, kidney cells were classified into ten cell types (CTs), including three subtypes of PTCs (CT1-3), glomerular epithelial cells (CT4), distal tubule cells (CT5), two subtypes of collecting-duct cells (CT6 and 7), and three types of immune cells (CT8-10). Overall, the 3D groups and 2DHK2_K groups were closer to the cluster of renal tubule cells (C1-6) and separated from the immune cells (CT8: NK-T, CT9: monocytes, and CT10: B cells), collecting duct intercalated cells (CT7), and 2DHK2_Z. For different subtypes of PTCs, the 3D cultured HK-2 cells were closer to the proximal straight tubule cells (CT3) than to the proximal convoluted tubule cells (CT1) and the PTCs without accurate classification (CT2). 

To further evaluate the 3D model, pairwise Pearson correlation tests were conducted on gene sets related to PTC selective markers or specific functions, including transport, drug-metabolizing enzymes, and metabolic enzymes (Figure 2D–G and Appendix A). (1) PTC marker genes were selected based on the curated set generated by Khundmiri et al. [13,14]. Consistent with the PCA results, 3D cultured HK-2 cells highly correlated with CT3 (r = 0.80) and displayed good correlation with CT1 (r = 0.70) or CT2 (r = 0.76), while the 2DHK2_K and 2DHK2_Z groups only correlated well with either CT3 (0.77) or CT1/2 (0.75), respectively (Figure 2D). (2) Transporter genes in two major classes, ATP-binding cassette transporters (ABC, ~50 members) and solute carriers (SLC, ~458 members) [32,33], were based on the literature and the Gene Nomenclature Committee (HGNC) website [34,35,36]. Many of these transporters expressed in PTCs have been reported and are important in drug transport, disposition, and drug–drug interactions. In this category, the 3D group correlated better with the native PTCs than the 2D groups did, particularly aligning well with CT1 (r = 0.81) (Figure 2E). Although the 2DHK2_K group is closer to the 3D group based on the overall gene expression (Figure 2C), it correlated less with native PTCs in terms of transporters. (3) A list of drug-metabolizing enzymes in the kidney was selected from the literature [28,37]. The 3D group highly correlated with CT2 (r = 0.84) and CT3 (r = 0.81), but moderately correlated with CT1 (r = 0.66). The 2DHK2_K group correlated better with CT3 (r = 0.79), whereas the association between the DHK2_Z group and all three PTC types was relatively weaker (Figure 2F). (4) Critical metabolism-related genes were selected from the literature [13,14]. Both the 3D and 2D groups correlated well with native PTCs (r > 90) (Figure 2G). Similar expression patterns from the 3D and 2D groups and three PTC clusters (PTC1-PTC3) were presented in the heatmap analysis (Figure 2H). This suggested that immortalized HK-2 cells still maintained a similar metabolic profile to native PTCs, except for small differences, such as glycolytic enzyme hexokinase (HK1) and fructolytic enzyme ketohexokinase (KHK).

### 2.3. Temporal Gene Expression Analysis of HK-2 Cells in 2D and 3D Culture

Proper transporter expression within an in vitro PTC model is essential for nephrotoxicity prediction. To validate the RNA-seq results, major drug transporters responsible for eliminating the top 200 prescribed drugs [37,38] were selected for further RT-qPCR analysis (Figure 3). These included transporters mediating basolateral uptake from the blood side, organic anion transporters (OAT1 and 3), and organic cation transporters (OCT1 and 2) [39,40], and transporters mediating apical efflux to the glomerular filtrate, multidrug and toxin extrusion (MATE1 and 2K), multidrug resistance-associated proteins (MRP2 and 4), and p-glycoproteins (MDR1/P-gp) [41]. Note that *OAT1*, *OAT 3*, *OCT2*, and *MRP2* expression were previously reported to be limited in 2D cultured HK-2 cells, while *MDR1* was expressed at relatively high levels [27]. Considering that each gene might express at particular time points, RT-qPCR analysis was performed from cells collected on Days 7, 14, 21, and 28. Average or peak expressions were compared. Most expression levels tested were elevated in HK-2 cells under 3D conditions (Figure 3A–I), except *OCT2*, *MRP4*, and *MDR1*. *OCT2* and *MRP4* exhibited comparable expression in both 2D and 3D conditions (Figure 3D,H), while the MDR1 expression was higher in the 2D conditions (Figure 3I). 

In addition to renal elimination, there are transporters mediating reabsorption, such as OAT4 and sodium–glucose cotransporter (SGLT) 1 and 2, located on the apical side of the PTCs. OAT4 is a bidirectional organic anion/dicarboxylate exchanger [42], and the SGLT family is involved in glucose uptake across the apical membrane [43]. OAT4 level was slightly higher in the 2D groups but not significantly, while *SGLT1* and *2* drastically increased in the 3D group (Figure 3J,L,M). Unlike other MRP members, MRP5 is located at the basolateral membrane and carries GSH and cyclic nucleotides [44,45]. *MRP5* expression was slightly higher in the 3D conditions (Figure 3K). As for the receptor-mediated endocytosis process, low-density lipoprotein receptor-related protein-2, megalin, was investigated. Megalin is responsible for tubule uptake of albumin [46]. The expression of *megalin* in HK-2 cells was significantly greater when cultured in 3D gelatin sponges than in 2D conditions (Figure 3N).

### 2.4. The Presence of Proximal Tubule Markers in the 2D vs. 3D Model

The epithelial and PTC phenotypes of 3D cultured HK-2 cells were further characterized. The epithelial markers Cytokeratin 8/18/19 belong to the cytoskeleton intermediate filament family and are mainly expressed in simple epithelial tissues [47]. The staining of Cytokeratin 8/18/19 was detected in both 2D and 3D groups as early as Day 7 and persisted through Day 28 (Figure 4A,B). γ-Glutamyl transpeptidase 1 (GGT1), a distinctive PTC marker, plays a role in the metabolism of glutathione. Elevated GGT1 activity has been associated with renal damage [48]. A more robust expression of GGT1 was present in HK-2 cells under 3D conditions (Figure 4C,D), suggesting 3D cultured HK-2 cells maintained the expression of epithelial and PTC markers. Selected transporter expression was also examined. Although OAT4 and MRP5 mRNA levels were slightly different between the 2D and 3D groups on Days 7, 14, and 21 (Figure 3J,K), their protein expression was more prominent under the 3D conditions (Figure 4G–J). Megalin protein expression was enhanced in 3D cultured HK-2 cells, consistent with the mRNA result (Figure 3F and Figure 4E). Further confirmation using quantitative western blot could further confirm these results. Figure 5 summarized RT-qPCR and protein staining results.

## 3. Discussion

The globally increasing prevalence of kidney diseases and the low identification rate (~2%) of drug nephrotoxicity in the preclinical studies [49] reinforce the need to develop a better predictive, reliable, but feasible model. In vitro renal cell models utilizing primary cells or immortalized cell lines have been widely used for nephrotoxicity evaluation and screening. Each displays advantages and disadvantages. Primary PTCs are more biologically relevant but are limited in availability, expanding capacity, rapid dedifferentiation, and variability [28,50,51]. Many immortalized or transformed PTC lines from different species, including commercial human PTCs, have been established. These lines are easy to maintain but were indicated to be less sensitive to known nephrotoxins, mainly due to altered expression of transporters after immortalization [52,53], culture methods, or even numbers of passages [54,55]. Moreover, the lack of comprehensive characterization of transcriptome profiling could also make decisions difficult regarding cell sources and culture methods. In this study, we intentionally selected a commonly used PTC line, HK-2, which has been reported to have no or low expression of key transporters and a low match to the native mouse PTC transcriptome [13,56]. HK-2 cells were seeded in a clinical-grade gelatin sponge to establish a simplified 3D human renal model and subject to systematic assessment.

### 3.1. HK-2 Cells Cultured in 3D Gelatin Sponges Displayed Long-Term Growth and Developed Tubule-like Structure

The proliferation rate and morphology of 2D and 3D cultured cells are usually different, depending on the cell lines [9,18]. Even the same cell line could proliferate at different rates in different types of 3D models. HK-2 cells cultured in the gelatin scaffold initially grew slower than those in the 2D culture. When the time was extended, the proliferation rate increased with sustained linear and stable growth for three weeks and resulted in higher cell numbers than those in the 2D culture (Figure 1). One explanation is that single cells under 3D conditions take a longer time to separate from the cluster of cells through proteolytic degradation than cells in the 2D culture system [57]. The 3D form of cells usually retains characteristic cell morphology and physiology of native tissue [58]. The tubular structure was observed in our 3D cultured HK-2 cells starting from Day 7, and the structures were still maintained until Day 21 (Figure 1). In contrast, we observed unstable growth of HK-2 cells with irregular sizes and shapes (oval, round, or elongated spindle) in the 2D condition (Figure 1 and Appendix A). The cells started to aggregate and form dome-shaped colonies after 7 days of culture. In all, the 3D HK-2 model in gelatin sponges presented greater structural support for tubule formation and a more biocompatible growth environment for the survival of HK-2 cells. 

### 3.2. HK-2 Cells Cultured in 3D Gelatin Sponges Recaptured Transcriptional Patterns of Human PTCs

Khundmiri et al. accomplished a large-scale transcriptome comparison between 14 different renal cell lines derived from 6 species and native rat kidney proximal tubules [13,14]. All cell lines were cultured in transwell-based 2D conditions. Based on their results, none of the lines fully matched the transcriptome of native rat PTCs, including HK-2 cells (a low-percentage match of 26%). With the newly reported transcriptome profile of human native kidney cells, we compared the expression profile of the 3D HK-2 model to that of native human PTCs and HK-2 cells in 2D culture to evaluate whether 3D cultured HK-2 cells are more representative of in vivo PTCs (Figure 2). Since active transport and metabolic pathways are essential for drug deposition and elimination and xenobiotics, we investigated the expression pattern of 2D and 3D cultured HK-2 and native PTCs in critical gene sets: overall PTC selective genes, transporters, drug-metabolizing enzymes, and energy and synthetic enzymes. Pairwise Pearson correlation tests were applied to 3DHK2, 2DHK2_K, and 2DHK2_Z groups, and the three clusters of human PTCs, corresponding to proximal convoluted tubules (CT1), the PTCs without accurate classification (CT2), and proximal straight tubules (CT3). 

In the comparison of 3DHK2 and two sets of 2D groups, the PCA showed that our 3DHK2 group is closer to the 2DHK2_K group but had a clear separation from 2DHK2_Z (Figure 2A–C), likely due to using different culture media. Both 3DHK2 and 2DHK2_K groups were cultured with KSF medium, while 2DHK2_Z was cultured with DMEM medium with 10% FBS (Table 1). The 3DHK2 group and 2DHK2_Z group were clustered with proximal straight tubule cells (CT3) and collecting-duct principal cells (CT6), closer to the other PTCs (CT1 and 2) and glomerular epithelial cells (CT4) but separated from the immune cells (CT8-10) and collecting-duct intercalated cells (Figure 3C). For PTC selective genes, a similar pattern was shown. The 3DHK2 group also correlated best with CT3 (0.80) and remained highly correlated with CT1, 2, and 6 (>0.7) but very low with CT7 and CT9. The 3D group correlated best with C2 and CT3 in genes related to drug metabolism and CT1 and CT2 in transporter genes. The expression of drug transporters and metabolizing enzymes is greater in the more distal parts of the proximal tubule [2,59]. In contrast, receptor-mediated endocytosis of filtered proteins is higher in the first part of the proximal tubule [60]. To confirm the presence of key transporters, selected gene expression was validated by RT-qPCR and immunostaining.

The expression profiles of energy and synthetic metabolic enzymes were unique in PTCs [14]. Both 2D and 3D groups correlated with PTCs very highly (Figure 2G), with higher expression levels of genes involved in gluconeogenesis (PCK1 and Slc37a4), lactate metabolism (LDHB), ammoniagenesis and arginine synthesis (Ass1), and the absence of glycogen synthase (Gys1). However, the expression of key glycolytic enzymes, hexokinase Hk1, was missing in the native PTCs, suggesting less usage of glucose for energy metabolism, but was present at high levels in cultured HK-2 cells. How this affects the metabolic machinery of HK-2 requires more investigation.

### 3.3. HK-2 Cells Cultured in 3D Gelatin Sponges Restored or Maintained More Key PTC Markers and Transporters

PTCs play important roles in the excretion and reabsorption of chemicals and xenobiotics through an array of transporters. Functional transporters may be lost in the in vitro model [61]. Jenkinson et al. reported the limitation of HK-2 cells as a renal model due to the lack of some important transporters, such as OAT1, OAT3, OCT2, and MRP2 [27]. Key transposers interacting with the top 200 prescribed drugs secreted by kidneys were examined by RT-qPCR at different time points.

Basolateral uptake transporters, OAT1, OAT3, OCT1, and OCT2, function in the clearance of organic anions, cations, uremic toxicity, and other drugs from the blood [40]. OCT1 can transport endogenous compounds such as choline and therapeutic drugs such as oxaliplatin [62]. Mice lacking OCT1 and OCT2 failed to secrete cisplatin, which led to severe renal damage [63]. OAT3 is involved in cellular uptake and secretion of therapeutic drugs and endogenous organic anions. A recent study showed that inhibition of OAT3 might reduce renal clearance and increase exposure to empagliflozin, which eventually leads to the suppression of SGLT2 [64]. Inhibition of OAT1 and OAT3 with some natural compounds, such as wedelolactone and wogonin, reduced cell viability [65]. Those studies emphasized the importance of the presence of these excretion transporters. In our 3D HK-2 model, weak or absent expression of OAT1, OAT3, and OCT1 in the 2D group was restored in the 3D gelatin sponge (Figure 3).

Apical efflux transporters, MDR1, MATE1, MATE2K, MRP2, and MRP4, mediate organic anions and cations for urine secretion. MDR1 and MRP2 and 4 belong to the ABC transporters family. MDR1 plays a significant role in the renal tubular secretion of organic cations such as cimetidine and some therapeutic drugs [41]. As reported [27], MDR1 was expressed at relatively high levels by 2D cultured HK-2 cells but was lowly expressed in 3D culture. MRP2 is responsible for the secretion of organic anions and drugs, such as cisplatin, into the lumen [66]. MRP4 is an organic anion transporter that can transport cAMP and antiviral drugs such as PMEA into the lumen [67]. Exclusive expression of MRP2 was observed in the 3D group, and comparable expression of MRP4 was observed between 2D and 3D groups. The MATE family has been identified as H^+^/organic cation antiporters. MATE1 mediates the efflux of cation compounds and zwitterionic drugs such as cephalexin [68]. Since cisplatin is secreted into urine by MATE1, inhibition of MATE1 by the anticancer drug pazopanib may lead to an increasing concentration of intracellular cisplatin, resulting in nephrotoxicity [69]. The function of MATE2 in the role of renal transporter remains unknown, but its variant MATE2-K secretes the cationic drug into the urine [69]. A recent study showed that MATE2 expression increased when cancer cells uptake the antidiabetic drug metformin in cancer therapy, indicating resistance to metformin [70]. Higher expression of MATE1 and MATE2 was exhibited in the 3D HK-2 model. The elevated expression of these transporters in our 3D HK-2 renal model may increase the sensitivity for screening drug-induced nephrotoxicity.

Additionally, in the gelatin-based 3D HK-2 kidney model, the reabsorption transporters (SGLT1, SGLT2) and endocytosis receptor (megalin) located in the apical membrane of HK-2 cells exhibited significantly higher expression levels than their 2D counterpart. Powell et al. demonstrated that SGLT1/SGLT2 double mutant mice failed to reabsorb the glucose back to PTCs and excreted it in the urine [71]. Although the levels of OAT4 and MRP5, an efflux transporter at the basolateral membrane, were similar, the immunostaining demonstrated higher protein expression in the 3D group, as well as megalin, GGT1 (PTC marker), and cytokeratin 8/18/19 (epithelial marker). In all, our results indicated that the increased expression of excretion transporters in the gelatin-based 3D HK-2 model makes it a more representative option for renal functional assays, drug screening, and nephrotoxicity applications. Notenboom et al. found an approximately 2-fold increase in MRP2 protein expression in the apical membrane [72]. Introducing OAT3 into the PTC cell line was used to increase the sensitive antiviral-induced cytotoxicity [73]. These results confirmed that increased expression of transporters to the membranes was beneficial in response to gentamicin exposure.

Taken together, the 3D model using gelatin scaffolds allowed HK-2 cells to enhance the expression of key transporters. More robust protein expression, such as megalin, OAT4, and GGT1, makes our 3D HK-2 renal model more applicable to native tissue. By minimally manipulating the culture microenvironment with a 3D gelatin scaffold, this simplified 3D renal model may provide a useful tool for preclinical drug testing. In conclusion, we established a more comparatively reliable PTC model with HK-2, recapitulating the in vivo transcriptome pattern and maintaining or restoring key renal markers and transporters, which contributes to the accurate prediction of nephrotoxicity studies. Furthermore, the detailed analysis of human cell lines will provide a systematic and comprehensive assessment of a variety of renal cell lines and culture methods.

## 4. Materials and Methods

### 4.1. HK-2 Cell Culture in 2D and 3D Models

The HK-2 cells (CRL-2190) were purchased from American Type Culture Collection (ATCC, Manassas, VA, USA) and were maintained in keratinocyte serum-free (KSF) medium (GIBCO, Dublin, Ireland) at 37 °C and 5% humidified CO_2_. Absorbable hemostatic gelatin sponges (Spongostan, Johnson & Johnson Medical, Somerville, NJ, USA) were chosen to generate the 3D model. The gelatin sponges were cut into a disc shape with a diameter of 8 mm and a thickness of 1 mm. Prior to use, the gelatin sponges were immersed in 75% ethanol, followed by three washes with PBS, and then soaked in PBS and then the medium for at least 1 day. An amount of 80 μL of cell suspension with two densities (10^5^ and 10^6^ cells/mL) was used for both the 3D culture and the respective 2D culture. In the 3D condition, the cell-loaded sponges were placed in 6-well low-attachment plates (Corning, NY, USA) to prevent cell adhesion to the plastic surface. For the 2D condition, HK-2 cells were cultured in 12-well culture dishes. The culture medium was changed every 2 or 3 days.

### 4.2. Cell Viability Assay

HK-2 cells were seeded with two initial seeding densities (10^5^ and 10^6^ cells/mL) for both 2D and 3D conditions and incubated with growth medium. The cells were harvested at 0, 3, 5, 7, 10, 14, 17, and 21 days for viability assessment using the WST-8 assay (Elabscience, Houston, TX, USA). After removing the culture medium, a diluted WST-8 solution (100 μL WST-8 reagent with 1 mL culture medium) was added to the cells. The mixture was then incubated for 1 h in a 5% CO_2_ humidified incubator at 37 °C. The color was determined by measuring the optical density (OD) at 450 nm and 655 nm. The assays were performed with three replicates for each experimental condition, and the cell viability of each replicate was measured three times. The data were presented as means ± standard error.

The live and dead assay was also performed using a LIVE/DEAD Viability/Cytotoxicity Kit (Thermo Fisher, Waltham, MA, USA) as follows. After being washed with PBS, HK-2 cells were mixed with the staining solution containing 1 μM calcein AM (CaAM, dye for live cells) and 10 μM ethidium homodimer-1 (EthD-1, dye for dead cells) and then incubated at 37 °C for 1 h. The samples were visualized and imaged under an inverted fluorescence microscope (Zeiss, Jena, Germany). For quantification, the number of live and dead cells in each view area (5X objective, 1.4 × 1 mm area) was counted. Three to five areas for each condition were measured. The data shown in bar plots were presented as means ± standard error.

### 4.3. Immunohistochemistry Analysis

For immunocytochemistry, the cell-loaded sponges and 2D cultures grown on cover slides were fixed with 10% paraformaldehyde for 1 h. Subsequently, the cells or sponges were washed twice with PBST (0.1% Tween 20) and then blocked with 10% normal goat serum and 1% bovine serum albumin (BSA) (Sigma, St. Louis, MO, USA) in PBS-T for 2 h at room temperature. After blocking and washing twice with PBST, the cells were immunostained with primary antibodies, anti-cytokeratin 8/18/19 (Abcam, Cambridge, UK, Ab41825; 1:100), anti-GGT1 (Abcam, Ab55138; 1:400), anti-OAT4 (Abcam, Ab76385; 1:100), anti-megalin (Abcam, Ab236244; 1:100) or anti-MRP5 (Abcam, Ab24107; 1:100) at 4 °C overnight, followed by washes and incubation with fluorescence-conjugated secondary antibodies at room temperature for 1 h. For F-actin staining, the cells were incubated with Alexa Fluor 568-conjugated phalloidin (Thermo Fisher Scientific, 1:500) for 1 h at room temperature. All cells were counterstained with DAPI. Images were captured using a confocal microscope TCS-SP8X (Leica, Wetzlar, Germany).

### 4.4. RNA Extraction for RNA-seq Analysis

HK-2 cells were cultured with an initial seeding density of 10^6^ cells/mL for the 3D condition and harvested on Day 14. Total RNA was extracted from three replicates of HK-2 cells using TRIzol (Thermo Fisher Scientific, Waltham, MA, USA), followed by DNase I (New England BIoLabs, Inc., Ipswich, MA, USA) treatment to eliminate genomic DNA. The RNA quality was evaluated by the A260/A280 ratio using a Biotek Epoch Microplate Reader (Agilent, Santa Clara, CA, USA). The RNA integrity was confirmed by 1.2% agarose gel electrophoresis.

### 4.5. Library Construction and Sequencing

RNA samples for high-throughput sequencing were subjected to an additional quality check using the Qubit2.0 with QubitTM RNA Broad Range Assay kit (Thermo Fisher Scientific, Waltham, MA, USA) and Caliper LabChip Analyzer (Caliper Life Sciences, Hopkinton, MA, USA). Libraries for sequencing were prepared using the KAPA mRNA HyperPrep Kit (Roche, Basel, Switzerland) according to the manufacturer’s protocol and sequenced on NovaSeq 6000 in PE150 mode. Raw sequencing reads were aligned to the human genome (GRCh38) by HISAT2 (v2.2.1) with the default setting [74]. Read counts and gene expression levels determined by RPKM [75] (reads per kilobase per million mapped reads) of each gene were calculated by the DEXSeq package (v1.46.0) [76] with customized R scripts. The RNA-Seq was carried out in three replicates for each condition. Reproducibility was confirmed by hierarchical clustering of all replicates, compared to the other datasets used in this study. All data will be available in Gene Expression Omnibus.

### 4.6. Transcriptome Comparison Analysis

To compare the transcriptome of 2D and 3D cultured HK-2 cells with that of native human PTCs, three published datasets were obtained from the GEO (Gene Expression Omnibus) database, including the following: (1) Human kidney scRNA-seq data (GSE131685) by Liao et al. [30]. The transcriptomic map was analyzed using a modified R script from Lioa et al. [30]. Cells were classified into ten cell types (CT1-10). The mean expression level for each cell type was calculated for further analysis. (2) Raw sequencing reads of 2D cultured HK-2 cells (GSE135441, 3 replicates) by Khundmiri et al. [13] and (3) Raw sequencing reads of 2D cultured HK-2 cells (GSE212681, 2 replicates) by Zhang et al. [31]. Most downstream analyses were performed using customized R script, unless specified. The sequencing reads were aligned to the human genome (GRCh38) using HISAT2, and RPKM for each gene was calculated using the DEXSeq package. To minimize technical differences, gene expression levels (RPKM) from all datasets were normalized using upper quartile normalization [77], followed by batch effect correction using the NOISeq package (v2.44.0) [78]. Pairwise Pearson correlation tests for the RPKM of selected genes in specific categories were performed using the psych package (v2.3.3) [79]. The resulting correlation plots were shown with either RPKM or log_2_ transformed RPKM. Correlation coefficient (r) and *p*-values were calculated. All *p*-values were smaller than 0.0001. Principle component analysis (PCA) based on the whole genome expression was also performed using the NOISeq package. Differentially expressed genes (DEGs) analysis between the 2D and 3D groups were analyzed and presented by volcano plots. Heatmaps of selected gene lists were generated to compare the expression levels among different culture conditions and human native PTCs using the ComplexHeatmap package (2.16.0) [80].

### 4.7. RNA Isolation and Quantitative Reverse Transcription Polymerase Chain Reaction (RT-qPCR)

To validate the RNA-seq results, RT-qPCR was performed on selected genes listed in Table 2. HK-2 cells were cultured with an initial seeding density of 10^5^ cells/mL for both 2D and 3D conditions and harvested on Day 7, 14, 21 and 28. RNA from three replicates of each condition was extracted and reverse-transcribed to cDNA using the iScript cDNA Synthesis kit (Bio-Rad, Hercules, CA, USA) following the manufacturer’s protocol. Quantitative PCR was performed using the iTaq Universal SYBR Green Supermix (Bio-Rad, Hercules, CA, USA) with the primers listed in Table 2. The expression levels of all target genes were normalized to the GAPDH value.

All qPCR data were presented in the relative expression levels (relative to GAPDH) and expressed as mean ± SE values. Statistical analysis was performed using the statistical software R (v4.3.1) within the R Studio software (v2023.06.1). The mean fold change between 2D and 3D groups was analyzed by all data points within each group regardless of culture days. The difference in peak expression between 2D and 3D groups was compared by the expression levels in the culture days with peak expression in each group. The difference between the two groups was analyzed through the permutation test in R with the default setting. Differences with *p*-value < 0.05 were considered statistically significant.

## Figures and Tables

**Figure 1 ijms-24-15547-f001:**
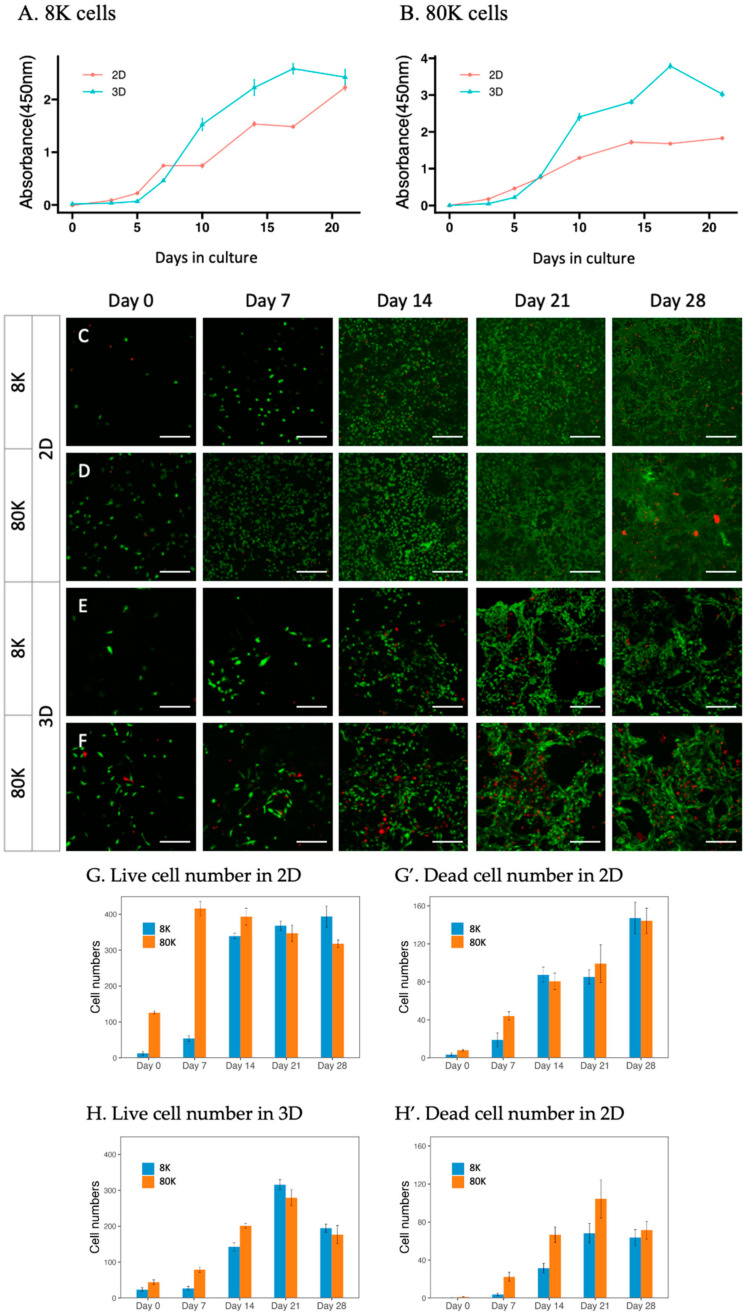
Cell growth and morphology in the 2D and 3D cultures. (**A**,**B**) WST-8 assay. Cell viability was characterized for 2D and 3D cultures with different seeding numbers, 8K (**A**) and 80K (**B**). After the addition of WST-8 reagent, the absorbance increase (450 nm) was detected after 1 h of incubation. The results were presented as mean ± standard error (ER) (*n* = 6). HK-2 cells in 3D culture initially showed a slower growth rate compared to those in the 2D culture, but the rate increased after 5 days of culture. (**C**–**F**) Live and dead staining of HK-2 cells in the 2D (**C**,**D**) and 3D (**E**,**F**) cultures with different initial seeding densities. Confocal images (scale bar: 250 μm) were taken using a fluorescent microscope on Days 7, 14, 21, and 28. Green: live cells, red: dead cells. Cells in 3D: a tubular-like structure was exhibited in 3D culture on Day 14 and was more prominent on Day 21 but was not observed in the 2D culture. (**G**,**H**) Quantitative comparison of live and dead cells in each indicated condition. The results were presented as mean ± standard error of the mean (SEM) (*n* = 6). (**I**–**K**) F-actin staining of HK-2 cells in the 2D (**I**) and 3D (**J**,**K**) culture observed at different time points by confocal microscopy. F-actin was stained with phalloidin (red), and the nuclei were counterstained with DAPI (blue). Magnifications: (**I**,**K**)—100×; (**J**)—200×. HK-2 cells in 3D culture revealed a tubule-like structure on Day 7 and persisted through Day 21.

**Figure 2 ijms-24-15547-f002:**
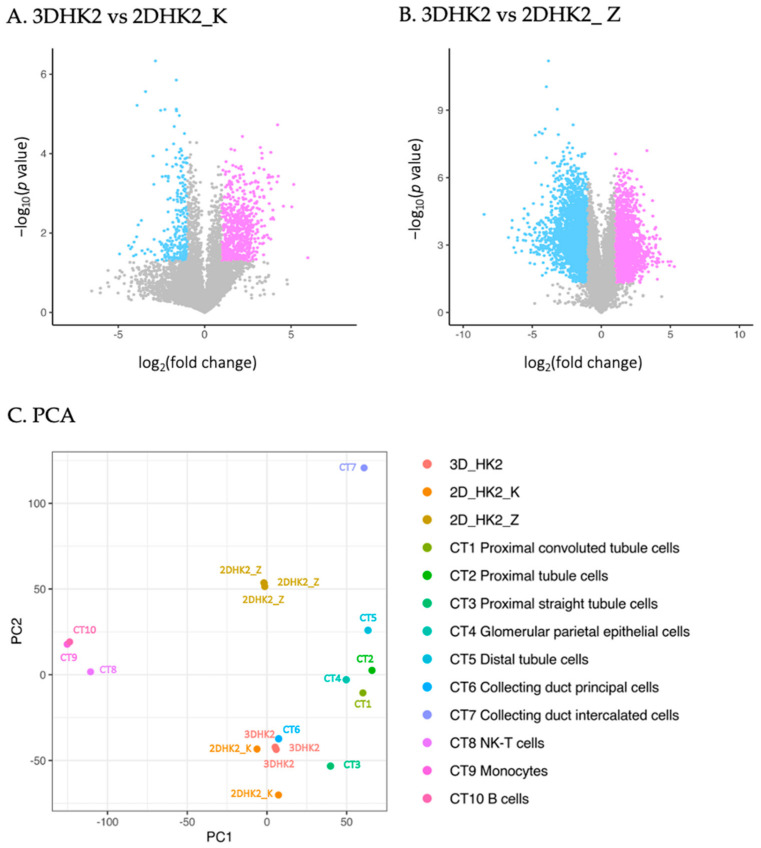
Transcriptome comparison of HK-2 cells between the 2D and 3D culture and native human renal cells. (**A**,**B**) The volcano plots (log_2_ fold change vs. −log_10_
*p*-value) showing DEGs between 3D and 2D cultured cells, 2DHK2_K and 2DHK2_Z, respectively. DEGs were identified if the fold change was ≥2.0 or ≤ 0.5 (up: pink or down: blue) and the *p*-value < 0.05. Non-DEGs are shown with gray dots. (**C**) PCA of gene expression levels in 2D cells (2DHK2_K and 2DHK2_Z), 3D cells (3DHK2), and 10 native human kidney cell types (CT) from published single RNA-seq data (CT1-10) based on the log_2_-transformed data. Cell types are as indicated. (**D**–**G**) Scatter plots showing the Pearson correlation tests of gene expression for PTC marker genes (**D**), SLC and ABC transporters (**E**), drug-metabolizing enzymes (**F**), and energy and synthetic enzymes (**G**) between the 3D or 2D culture cells and native PTCs. Correlation coefficient (r) is as indicated. All *p*-values are smaller than 0.0001. (**H**) Heatmap showing DEGs for selective metabolism-related genes between the 2D and 3D cultures and native PTCs.

**Figure 3 ijms-24-15547-f003:**
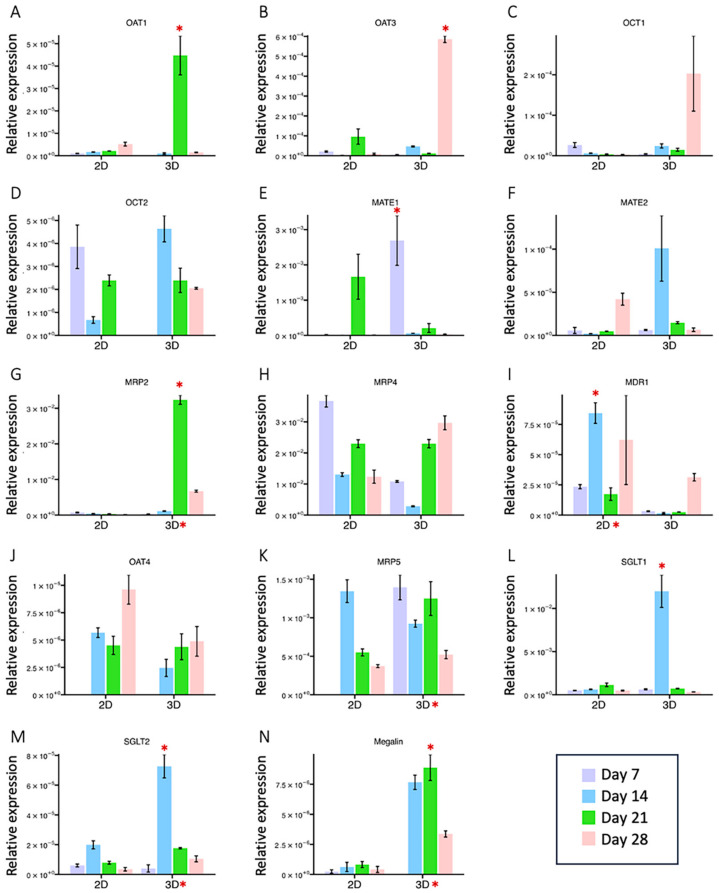
Comparison of gene expression levels of HK-2 cells between the 2D and 3D cultures. (**A**–**D**) Transporters mediating basolateral uptake. (**E**–**I**) Transporters mediating apical efflux. (**J**–**M**) Transporters mediating reabsorption. (**N**) Receptor mediating endocytosis. Relative expression levels of selected transporters (as indicated) were evaluated by RT-qPCR analysis. Mean fold change and peak level fold change (3D value/2D value) with their respective *p*-values were calculated and listed in Appendix A. Significant differences (FC > 1.5 or <0.67, *p*-value < 0.05) of mean fold change are indicated by an asterisk next to the 2D or 3D groups, and significant differences of peak level are indicated by an asterisk above the bar.

**Figure 4 ijms-24-15547-f004:**
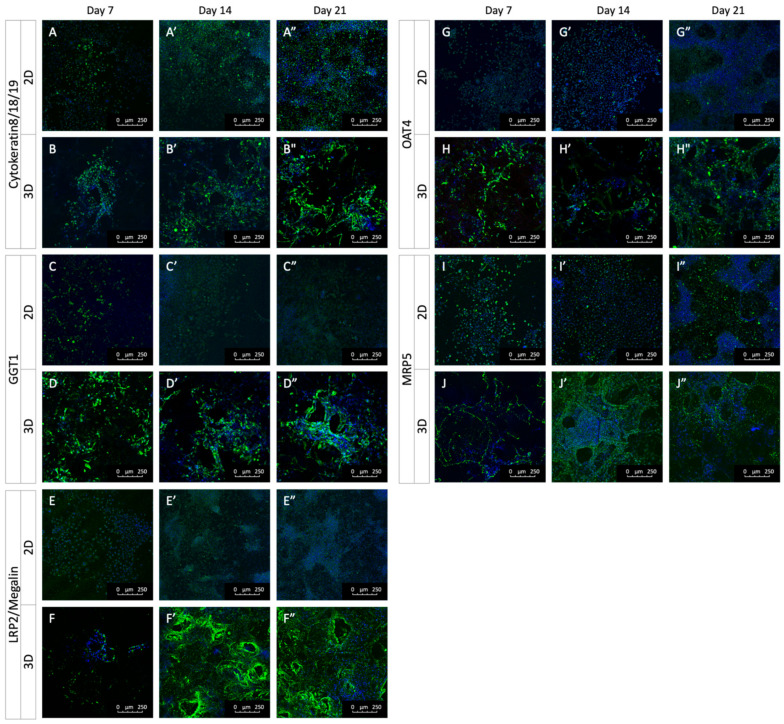
Comparison of the epithelial markers and transporters in the 2D and 3D HK-2 cell cultures. Expression of epithelial cell markers, cytokeratin 8/18/19 (**A**,**B**), PTC marker GGT1 (**C**,**D**), endocytosis receptor, megalin (**E**,**F**) and transporters, OAT4 (**G**,**H**), and MRP5 (**I**,**J**) at different timepoints as indicated.

**Figure 5 ijms-24-15547-f005:**
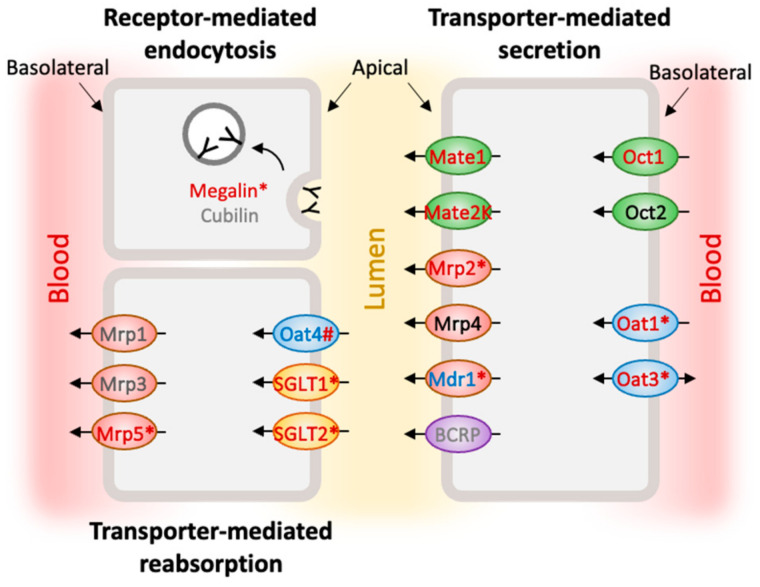
Summary of key carriers and drug transporters in cultured PTCs. Text in red: upregulated mRNA expression in the 3D group. Text in blue: downregulated mRNA expression in the 3D group. Text in black: similar mRNA expression between the 2D and 3D groups. Text in gray: key transporters that were not tested by RT-qPCR. * Indicates significant differences. # Indicates upregulated protein levels in the 3D group.

**Table 1 ijms-24-15547-t001:** Data sources, cell or tissue sources, and culture conditions of RNA-seq data used for comparison.

Data Sources	Cells	Culture Method	Medium	Culture Period
**From this study (3DHK2)**	HK-2	3D: Gelatin sponge	KSF Medium	14 days
**Khundmiri et al. [13] (2DHK2_K)**	HK-2	2D: Transwell	KSF Medium	95–98% confluence
**Zhang et al. [31] (2DHK2_Z)**	HK-2	2D: 25 cm^2^ flask	DMEM medium with 10% FBS	NS
**Liao et al. [30] (scRNA-seq_L)**	Human kidney cells	NA	NA	NA

NS: not specified; NA: not applicable.

**Table 2 ijms-24-15547-t002:** Sequences of the primer pairs used for real-time quantitative reverse transcription polymerase chain reaction.

Gene	Sequence or TaqMan^®^ Assay IDs
*GAPDH*	Forward primer 5′-GGCGCTGAGTACGTCGTGGAG-3′ Reverse primer 5′-ATTGCTGATGATCTTGAGGCTGTTG-3′
*MATE1*	Forward primer 5′-TGATCAGGAACACCATCAGC-3′ Reverse primer 5′-GAGGCCACCCTTGAGGTC-3′
*MATE2*	Forward primer 5′-TGCTTCCCAGTTCCTCTCAG-3′ Reverse primer 5′-GAAGATGTCATTGCCCTGGT-3′
*MDR1*	Forward primer 5′-CCTAGGAGTACTCACTTCAGGA-3′ Reverse primer 5′-AAGATCCATTCCGACCTCGC-3′
*megalin*	Forward primer 5′-TGGATGTGAAAGCGGTCCTC-3′ Reverse primer 5′-ACTCAACACAGGTACGGCTG-3′
*MRP2*	Forward primer 5′-ACGCAGTCCAGGAATCATGC-3′ Reverse primer 5′-AAAACCAGGAGCCATGTGCC-3′
*MRP4*	Forward primer 5′-GTGGCCGTGATTCCTTGGAT-3′ Reverse primer 5′-GGCATCCAGAGTTTTTGCCAG-3′
*MRP5*	Forward primer 5′-CTGAAGCCCATCCGGACTAC-3′ Reverse primer 5′-CACCATGAAGGCTGGTCCAC-3′
*OAT1*	Forward primer 5′-TGTCATCAACTCCCTGGGTCG-3′ Reverse primer 5′-CCCCAGCACAGCAAGAGAGGT-3′
*OAT2*	Forward primer 5′-AGCCTACGTGAGTACCCTGG-3′ Reverse primer 5′-CACTCCAGCTCCAGTGGC-3′
*OAT3*	Forward primer 5′-CACGAGCCCTCCAATCAGTA-3′ Reverse primer 5′-CTGGGTCTACAACAGCACCA-3′
*OAT4*	Hs00945829_m1
*OCT1*	Forward primer 5′-CCCCTCATTTTGTTTGCGGT-3′ Reverse primer 5′-TTTCTCCCAAGGTTCTCGGC-3′
*OCT2*	Forward primer 5′-GATAGTCTGCCTGGTCAATGCTG-3′ Reverse primer 5′-GAGCCGGTAGACCAGGAATG-3′
*SGLT1*	Forward primer 5′-TGGCCACTTCCAATGTTACT-3′ Reverse primer 5′-GGGACTGTTGGAGGCTTCTT-3′
*SGLT2*	Forward primer 5′-GTCATCGCGCTTCTGGGCAT-3′ Reverse primer 5′-GGAAGGCGTAACCCATGAGGAT-3′

## Data Availability

Data will be made available on request.

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
