# Peer review of "Delivery and Transcriptome Assessment of an In Vitro Three-Dimensional Proximal Tubule Model Established by Human Kidney 2 Cells in Clinical Gelatin Sponges"

_ijms, 2023, doi:10.3390/ijms242115547_

Round 1
Reviewer 1 Report
Hi Authors,
The article on 'The Delivery and Transcriptome Assessment of An in vitro Three-Dimensional Proximal Tubule Model established by Human kidney 2 Cells in Clinical Gelatin Sponges' is written well and the methodology with appropriate experiments.
1. The abstract and introduction are apt.
2. Line 102 - 8000 or 80,000 of HK-2 cells were seeded. Any rationale? why two densities?
3. Figure 1. Addition of day 0 or 1 would be appreciated. Labelling is missing?
4. Figure 3. The colors on graphs look similar. Change of different distinct colors for graphs is required.
5. Figure 5. The schematic looks great!
Minor revision is needed!
Reviewer 2 Report
In their well-designed study the Authors demonstrated that developed by them 3D PTC model matches native tubular cells much better than commonly used 2D models. The match refers to a big number of genes and respective proteins that are crucial for PTC functions but are lost in 2D culture. Therefore, the 3D model presented in this study seems to be a good alternative for the in vitro studies.
Reviewer’s comments:
1. Line 30 : keywords are missing
2. Lines 35-37: leading causes of kidney impairment include first of all civilization diseases, i.e. diabetes and hypertension
3. Figure 1; the graphs are too small making it difficult to read. Also, the pictures showing live and dead cells are very difficult to assess
4. Figure 1 caption refers to A, B, C, D…etc but only A and B graphs are marked properly. It is extremely difficult to match certain pictures and description below.
5. Fig 3: pronounced differences between 2D and 3D expressing certain genes are obvious but they should be confirmed by statistical analysis (P values) or by respective information in Figure caption. This would make it easier to read than to find it in the Methods paragraph. Explanation (graph legend) is missing, it is not clear what do different bar colors represent
6. Fig 4: Is it DAPI that stains blue? This should be mentioned in figure caption. However, blue areas are vey big, much bigger than the cell size, so it seems unlikely that it is nuclear staining.
7. There is a discrepancy between RT-PCR (Fig 3) and IF staining (Fig 4) results. OAT4 mRNA expression is higher in 2D than in 3D , in contrast to immunostaining that is poor (if any) in 2D,while it is quite nice in 3D. In general, quality of images is low for all tested markers in 2D cells. The pictures suggest that 2D cells don’t express any of tested proteins. Is that true? Quantitative Western blot or flow cytometry analysis would be a convincing proof.
8. Fig 5: OAT4 is presented in red which, according to the figure caption means that this transporter was upregulated in D3 cells. However, Fig 3 indicates that expression of OAT4 is lower in D3 than in D2 group
The manuscript needs English proofreading, since numerous typos, grammar and syntax errors, make it sometimes difficult to understand
Round 2
Reviewer 2 Report
The Authors have carefully revised their manuscript making it fully acceptable